# SurfelSoup: Probabilistic G-SurfelTree for Learned Point Cloud Geometry Compression

## Abstract

This paper presents SurfelSoup, the first end-to-end learned surface-based framework for dense point cloud geometry compression, with surface-structured primitives for representation. It proposes a probabilistic and differentiable surface representation, G-Surfel, which models local point occupancies using a bounded generalized Gaussian distribution. We further introduce G-SurfelTree, an octree-like hierarchy, where a decision module adaptively terminates the tree subdivision for rate-distortion optimal G-Surfel granularity selection. This formulation avoids redundant point-wise compression in smooth regions and produces compact yet smooth surface reconstructions. Experimental results under the MPEG common test condition show consistent gain on dense geometry compression over voxel-based baselines and MPEG standard G-PCC-GesTM-TriSoup, while providing visually superior reconstructions with smooth and coherent surface structures.

## 1 Introduction

Point clouds, which represent the 3D structures using colored points, have emerged as a crucial 3D data representation in the field of immersive media, autonomous driving, and virtual/augmented reality (Huang et al., 2025; Jiang et al., 2025), where compression is necessary to reduce the storage and transmission costs. To this end, the Moving Picture Experts Group (MPEG) established two standards for point cloud compression (PCC): video-based PCC (V-PCC) (Schwarz et al., 2018) and geometry-based PCC (G-PCC) (Zhang et al., 2024). Beyond these standards, numerous learning-based methods have emerged (Quach et al., 2020; Wang et al., 2021b;a; 2023; 2024; Que et al., 2021; Fan et al., 2022; You et al., 2025). Despite the great advancements, the fact that points are unevenly distributed in the 3D space poses significant challenges to the design of efficient PCC schemes.

To tackle the above problem, research in point cloud compression focuses on regularizing the point cloud geometry with different 3D representations. G-PCC-Octree uses the octree structure, which transfers the non-uniform points into a tree structure. To exploit the surface structure in dense point clouds, G-PCC-TriSoup represents a point cloud as a set of triangles, significantly outperforming G-PCC-Octree. Meanwhile, several learned point cloud compression methods (Wang et al., 2021b; 2022; 2024; Wang & Gao, 2025) have been proposed to leverage sparse convolution (Choy et al., 2019) on the voxelized point clouds. Although these learned approaches improve compression efficiency, they remain fundamentally tied to voxelized representations and octree-based structures, with redundant voxel divisions in smooth regions ignoring the underlying surface structure, leading to inherently less efficient compression. In addition, voxel-based methods often struggle to preserve continuous surfaces and tend to produce artifacts such as gaps at low bit rates.

To this end, we break from the voxel-based paradigm and introduce SurfelSoup. To the best of our knowledge, this is the first end-to-end surface-based framework for point cloud compression, with surface primitives for reconstruction. Rather than encoding geometry as voxels, SurfelSoup models it as a composition of probabilistic surfaces, termed G-Surfels, organized in an octree-like structure called G-SurfelTree (Fig. 1). Each G-Surfel describes voxel occupancy likelihoods within an octree node using a differentiable bounded 3D generalized Gaussian distribution, enabling end-to-end optimization. A decision module adaptively decides whether to terminate a tree node with a G-Surfel, or to further split it into children nodes, based on the desired rate-distortion trade-off. Experiments show that, compared with voxel-based methods, SurfelSoup achieves greater coding efficiency and qualitatively superior reconstructions with smooth and coherent surface structures.

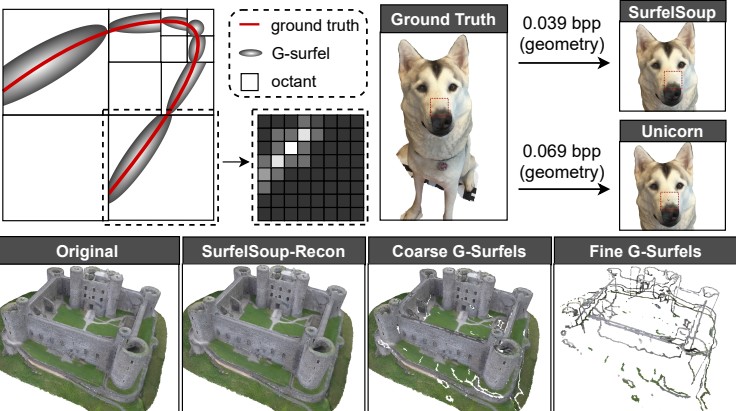

Figure 1: An illustration of SurfelSoup. Top Left: a toy 2D example for the G-SurfelTree structure. Top Right: Example reconstruction comparison of a 3D Husky. Bottom: Example of G-SurfelTree's multi-layer reconstruction.

Our main contributions include:

- **The first end-to-end surfaced-based point cloud geometry compression framework, SurfelSoup**, leveraging a probabilistic surface representation, termed **G-Surfel**. It models the voxel occupancies in an octree node by a bounded 3D generalized Gaussian (GG) function, enabling the use of a differentiable distortion term in the loss function for training.

- **A G-SurfelTree hierarchy for adaptive G-Surfel granularity assignment**. Instead of using a pre-defined tree structure, G-SurfelTree makes rate-distortion optimal assignment of G-Surfels at different octree levels. This is achieved through the proposed G-SurfelTree decision module and a corresponding formulation of the total expected rate and distortion.

- **Validations according to the MPEG common test condition (CTC) for AI-PCC**. Our experiment results show that SurfelSoup outperforms the previous state-of-the-art voxel-based method Unicorn-geometry (Wang et al., 2024) and other voxel-based approaches, as well as the surface-based G-PCC-TriSoup, for dense point cloud geometry compression.

## 2 RELATED WORK

### 2.1 RULE-BASED POINT CLOUD COMPRESSION

The Moving Picture Experts Group (MPEG) specifies video-based PCC (V-PCC) and geometry-based PCC (G-PCC). V-PCC projects 3D point clouds onto 2D. G-PCC encodes the geometry directly in 3D using regularized data structures. G-PCC-Octree represents the geometry as an occupancy octree, while G-PCC-TriSoup approximates points under each octree node with triangles at a chosen octree level. TriSoup has been shown to outperform G-PCC-Octree (Quach et al., 2020) for dense point clouds, as it encodes surfaces with far fewer parameters than the voxel occupancies.

### 2.2 LEARNING-BASED POINT CLOUD COMPRESSION

Early learning-based point cloud compression methods borrow the idea of end-to-end entropy models (Ballé et al., 2017; 2018; Minnen et al., 2018) and adopt 3D convolutions to process volumetric inputs (Quach et al., 2020; Wang et al., 2021b; Nguyen et al., 2021). However, the excessive computational complexity of dense 3D convolutions requires pre-partitioning of the input (Quach et al., 2020; Wang et al., 2021b), resulting in long encoding/decoding times and degraded efficiency. To address this, PCGCv2 (Wang et al., 2021a) first introduces 3D sparse convolutions (Choy et al., 2019) into point cloud compression, greatly reducing computational cost and eliminating the need for pre-partitioning. Building upon this, SparsePCGC (Wang et al., 2022) proposes a unified $N$-stage Sparse Octree Probability Aggregation (SOPA) module for both lossless and lossy compression, which auto-regressively models octant occupancy distribution in a fixed Morton order.

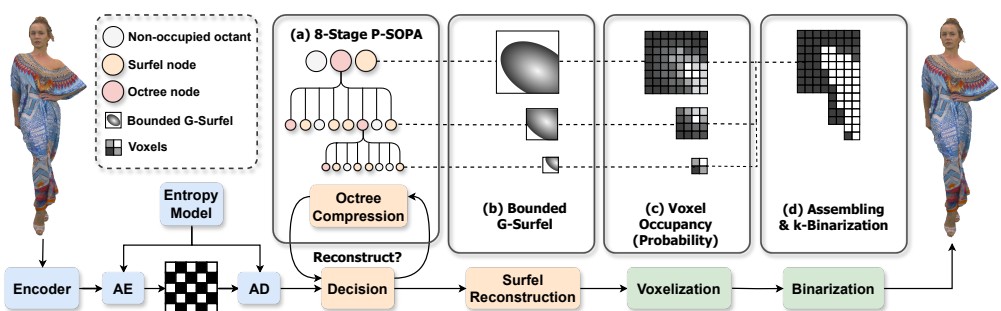

Figure 2: The overall architecture of SurfelSoup.

ViewPCGC (Zheng et al., 2024) further introduces view information into point cloud compression. More recently, UniPCGC (Wang & Gao, 2025) introduces a Variable Rate and Complexity Module that enables adaptive rate control, further improving over SparsePCGC. Unicorn (Wang et al., 2024) proposes a unified architecture for both geometry and attribute compression, which incorporates a Neighborhood Point Attention (NPA) module for geometry compression. There are also works that focus on LiDAR point clouds (Huang et al., 2020; Biswas et al., 2020; Fu et al., 2022; You et al., 2025), exploiting sparsity and range-image structures. These designs are effective for LiDAR but do not generalize well to dense point clouds (Wang et al., 2024), which are the focus of our work.

### 2.3 SURFACE REPRESENTATION FOR POINT CLOUDS

Representing 3D surfaces for reconstruction, processing, and rendering has been a fundamental research problem. To this end, both implicit (Park et al., 2019; Fridovich-Keil et al., 2022; Mildenhall et al., 2021; Chibane et al., 2020) and explicit surface representations (Pfister et al., 2000; Dai et al., 2024; Gao et al., 2023; Aliev et al., 2020) have been developed. Notably, surfel (Pfister et al., 2000) proposes to represent a surface with isolated surface elements without storing the connectivity, hence achieving efficient rendering and processing flexibility. Later works introduce the surface splatting (Zwicker et al., 2001; Kobbelt & Botsch, 2004) techniques and non-square shapes (Botsch et al., 2005) to address the visibility issues in rendering. By designing the primitives with differentiability, surfels and more generally 3D Gaussians have been shown to facilitate 3D reconstruction from multi-view images and novel view rending Kerbl et al. (2023); Dai et al. (2024).

Our work is inspired by the work (Pfister et al., 2000) with surfels as primitives for surface representation, but we focus on the compression of point clouds which represent an underlying surface, and our goal is to reduce the bit-rate while maintaining the surface representation accuracy. Our work is orthogonal to 3D Gaussian splatting, as well as other implicit or explicit surface representations, where the goal is to reconstruct 3D structures from multi-view images. Our work is most related to TeSO (Hu et al., 2025), where the 3D scene is described by cube-bounded textured surfels organized on an octree, with the geometry of each surfel parameterized by a normal vector, an octree bounding box, a center, and a radius. However, the TeSO is first constructed based on a hand-crafted criterion for determining whether to terminate an octree node with a surfel, and then the TeSO parameters are compressed. Such an approach fails to achieve rate-distortion optimality.

## 3 METHODOLOGY

### 3.1 OVERVIEW

The architecture of SurfelSoup is illustrated in Figure 2. The input point cloud $\mathbf{p}$ is represented as a sparse tensor $\mathbf{p} = [\mathbf{p}_C, \mathbf{p}_A]$, where $\mathbf{p}_C$ denotes the 3D coordinates and $\mathbf{p}_A$ the associated attributes. For geometry compression, $\mathbf{p}_A = \mathbf{1}$ to indicate occupancy. The encoder module (Sec. 3.2) maps $\mathbf{p}$ into latent $\mathbf{f}^L$ via sparse convolution, where the superscript "$L$" denotes $L$ stages of octree-based downsampling. $\mathbf{f}^L$ is quantized, arithmetically coded and decoded (AE/AD) to yield $\hat{\mathbf{f}}^L$. For each non-empty node $\mathbf{u}_i^L$ with latent $\hat{\mathbf{f}}_i^L$, the decision module (Sec. 3.4) selects between:

1. reconstructing a G-Surfel via the surfel reconstruction module (Sec. 3.3), or

2. compressing the occupancy information of the eight child octants of the current node using the octree compression module (Sec. 3.5).

The above process is then recursively applied on non-empty octants generated by Decision 2. The recursive process yields a tree-structured representation, termed the G-SurfelTree, where a node is a leaf if and only if it reconstructs a G-Surfel. At octree layer $l \in [0, L]$, we denote by $\mathbf{s}^l$ the set of nodes following Decision 1 (surfel nodes), and $\mathbf{o}^l$ the set of nodes following Decision 2 (octree nodes). Finally, each surfel node $\mathbf{s}_i^l$ is converted into a local point set with assigned occupancy likelihoods based on the decoded G-Surfel parameters via the voxelization module, and all such sets are assembled and reconstructed into a point cloud through the binarization module.

## 3.2 Encoder & Entropy Coding

The encoder module maps the input point cloud $\mathbf{p}$ into a latent $\mathbf{f}^L$. It consists of $L$ downsampling stages following the design of (Wang et al., 2024), implemented via sparse convolutions. The latent representation is decomposed as $\mathbf{f}^L = \left[ \mathbf{f}_C^L, \mathbf{f}_A^L \right]$, where the geometric coordinates $\mathbf{f}_C^L$ are losslessly compressed using the G-PCC-Octree, while the attributes $\mathbf{f}_A^L$ (containing the latent features) are quantized and entropy coded using the hyperprior entropy model (Ballé et al., 2018). The reconstructed attribute latent $\hat{\mathbf{f}}^L$ is then upsampled to each octree level $l$ via sparse transposed convolutions (See Appendix), yielding $\hat{\mathbf{f}}^l, l = 1, 2, \ldots, L-1$ for subsequent G-SurfelTree construction.

## 3.3 G-Surfel Representation

Our G-Surfel is formulated as a probabilistic soft plane, modeling the spatially varying occupancy likelihood for voxels in an occupied octree node using a 3D generalized Gaussian (GG) function:

$$P(\mathbf{x}) = \exp\left( -\frac{1}{2} r(\mathbf{x}, \boldsymbol{\Sigma})^{2\beta} \right), \quad r(\mathbf{x}, \boldsymbol{\Sigma}) = \sqrt{(\mathbf{x} - \mu)^T \boldsymbol{\Sigma}^{-1} (\mathbf{x} - \mu)}, \tag{1}$$

where $\mathbf{x} \in \mathbb{R}^{\mathbf{3}}$ denotes any location in 3D space.

We follow Gaussian Splatting (Kerbl et al., 2023)'s settings to describe the 3D GG function. For each surfel node $\mathbf{s}_i^l$, the surfel reconstruction network generates 11 parameters: the mean $\mu_i^l \in \mathbb{R}^3$, standard deviation $\sigma_i^l \in \mathbb{R}^3$, quaternion $\mathbf{q}_i^l \in \mathbb{R}^4$, and shape coefficient $\beta_i^l \in \mathbb{R}$, based on $\hat{\mathbf{f}}_i^l$.

Note that the 3D GG function reduces to a 2D function describing a planar surfel when one of its rotated axes has a very small variance. This axis is the normal vector of the surfel, and the other two axes define the two canonical axes on the surfel (which can be used to define the texture patch in the TeSO representation). The $\sigma_i$ along each plane axis defines the spread of the surfel. The parameter $\beta$ controls the sharpness of the drop off beyond the respective $\sigma$ in each axis. The GG function reduces to the Gaussian function when $\beta = 2$, which have slower drop-offs.

**Bounding Box.** To enable G-Surfels to exploit the boundaries of the bounding boxes to construct complex geometries, we restrict the influence of each G-Surfel to the bounding box defined by the octree, *i.e.* the coverage region of $\mathbf{s}_i^l$ with side length $2^l$, as shown in Fig. 2 (b).

**Voxelization and distortion incurred by G-surfel approximation.** To recover the discretized point cloud from the probabilistic G-Surfel, the voxelization module computes the occupancy likelihood $P(\mathbf{v}_{i,k}^l)$ of any $\mathbf{v}_{i,k}^l \in \mathcal{V}_i^l$, the set of voxels (integer points) inside the bounding box of $\mathbf{s}_i^l$. The distortion of $\mathbf{s}_i^l$ represented by a G-surfel is measured by the binary cross-entropy (BCE):

$$\mathcal{D}(\mathbf{s}_i^l) = -\sum_{\mathbf{v}_{i,k}^l \in \mathcal{V}_i^l} \left[ Q(\mathbf{v}_{i,k}^l) \log\left( P(\mathbf{v}_{i,k}^l) \right) + \left( 1 - Q(\mathbf{v}_{i,k}^l) \right) \log\left( 1 - P(\mathbf{v}_{i,k}^l) \right) \right], \tag{2}$$

where $Q(\mathbf{v}_{i,k}^l)$ denotes the ground truth occupancy of $\mathbf{v}_{i,k}^l$.

**Binarization.** At the inference time, the binarization module converts the continuous occupancy probabilities into binary occupancy codes for point cloud reconstruction. Specifically, all $\mathcal{V}_i^l$ are assembled into a global set $\mathcal{V}$, among which the top $\rho N$ voxels with the highest occupancy probabilities are selected as the reconstructed point cloud, where $N$ denotes the number of points in the original point cloud, and $\rho$ is a user-defined scaling factor (set to 1 in our experiment).

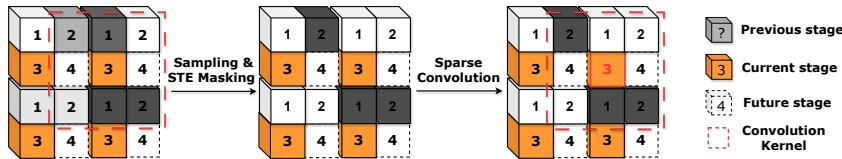

Figure 3: A 2D example to explain P-SOPA. The octants labeled 1 and 2 have been coded, with gray intensities indicating the existence probabilities (the darker the higher probability). Octants labeled 3 are being coded, and octants labeled 4 are yet to be coded and unseen. In the middle figure, some octants labeled 1 and 2 were randomly masked out (set as white). Real P-SOPA is done on 3D.

| **Algorithm 1** G-SurfelTree Construction | **Algorithm 2** P-SOPA |
|---|---|
| **Require:** Latent $\hat{\mathbf{f}}_i^l$, octants $\mathcal{O}_i^l$ | **Require:** Latent $\hat{\mathbf{f}}_i^l$, octants $\mathcal{O}_i^l$ |
| 1: **function** CONSTRUCTTREE($\hat{\mathbf{f}}_i^l$) | 1: **function** P-SOPA($\hat{\mathbf{f}}_i^l$) |
| 2:    $\tilde{p}_i^l, \tilde{q}_i^l \leftarrow$ DECISIONMODULE($\hat{\mathbf{f}}_i^l$) | 2:    **for** $j = 0$ **to** 7 **do** |
| 3:    $\mathcal{D}(\mathbf{s}_i^l) \leftarrow$ SURFELRECON($\hat{\mathbf{f}}_i^l$) | 3:      $\mathcal{P}_{[:j-1]}^l \leftarrow \{\tilde{p}_{i'}^l | \forall i'; j' \in [: j-1]\}$ |
| 4:    $\mathcal{B}(\mathbf{o}_i^l), \hat{\mathbf{f}}_{i,[0:7]}^l \leftarrow$ P-SOPA($\hat{\mathbf{f}}_i^l$) | 4:      $m_{[:j-1]}^l \sim Bernoulli(\mathcal{P}_{[:j-1]}^l)$ |
| 5:    **for** $j = 0$ **to** 7 **do** | 5:      $\bar{m}_{[:j-1]}^l = $ STE($m_{[:j-1]}^l$) |
| 6:      **if** $(\mathcal{O}_{i,j}^l = 1) \wedge (\text{train} \vee \tilde{q}_i^l < \epsilon)$ **then** | 6:      $[\theta, \hat{\mathbf{f}}]_{i,j}^l \leftarrow$ SOPA($\hat{\mathbf{f}}_i^l, \mathcal{O}_{[:j-1]}^l, \bar{m}_{[:j-1]}^l$) |
| 7:        CONSTRUCTTREE($\hat{\mathbf{f}}_{i,j}^l$) | 7:    **end for** |
| 8:      **end if** | 8:    $\mathcal{B}(\mathbf{o}_i^l) \leftarrow$ ARITHMCODER($\theta_i^l, \mathcal{O}_i^l$) |
| 9:    **end for** | 9:    **return** $\mathcal{B}(\mathbf{o}_i^l), \hat{\mathbf{f}}_{i,[0:7]}^l$ |
| 10: **end function** | 10: **end function** |

### 3.4 G-SURFELTREE DECISION

To adaptively decide the optimal granularity of constructing a G-Surfel, the decision module decides whether an occupied node should terminate as a surfel or be further split. To enable the joint training of the decision module with other modules, we replace the binary decisions with differentiable probabilistic assignments during training. However, computing the **marginal probability** that a node $\mathbf{u}_i^l$ is an octree node (denoted as $\tilde{p}_i^l$) or a surfel node ($\tilde{q}_i^l$) is intractable, since a node exists only if its parent is an octree node requiring further splitting. Therefore, the decision module generates a **conditional probability** $p_i^l$ that the node $\mathbf{u}_i^l$ is an octree node from its latent $\hat{\mathbf{f}}_i^l$, conditioned on either 1) its parent is also an octree node with probability $\tilde{p}_{\pi(i,l)}^{l+1}$; or 2) it is a top-level node ($l = L$). The probabilities $\tilde{p}_i^l$ and $\tilde{q}_i^l$ are therefore recursively computed as:

$$\tilde{p}_i^l = p_i^l \tilde{p}_{\pi(i,l)}^{l+1}, \quad \tilde{p}_i^L = p_i^L; \quad \tilde{q}_i^l = (1 - p_i^l)\tilde{p}_{\pi(i,l)}^{l+1}, \quad \tilde{q}_i^L = 1 - p_i^L, \tag{3}$$

where $\pi(i, l)$ denotes the index of the parent of $\mathbf{u}_i^l$.

To make the decisions differentiable while close to binary during training, we adopt the Gumbel–Sigmoid reparameterization (Jang et al., 2017), providing a continuous relaxation of sampling.

The overall algorithm of G-SurfelTree construction using our decision module is shown in Alg. 1. During evaluation, nodes with $\tilde{q}_i^l \geq 1 - \epsilon$ are classified as surfel nodes $\mathbf{s}_i^l$ and the recursion terminates. During training, however, no hard termination is applied; each node is represented in a probabilistic manner, allowing all branches to remain active and differentiable.

### 3.5 OCTREE NODE OCCUPANCY COMPRESSION: P-SOPA

To compress the occupancy of the octants of an octree node, we adopt the 8-Stage SOPA (Wang et al., 2022), which employs a local auto-regressive structure conditioned on the already decoded octants. Define the octants of an octree node $\mathbf{o}_i^l$ as $\mathcal{O}_i^l = \{\mathcal{O}_{i,j}^l \mid j \in \{0, 1, \cdots, 7\}\}$, where $\mathcal{O}_{i,j}^l \in \{0, 1\}$ denotes the $j$-th octant in Morton order. For each octree layer $l$, octants with the same index $j$, i.e. $\mathcal{O}_{[j]}^l = \{\mathcal{O}_{i,j}^l | \forall i\}$, are grouped together and coded in parallel. Specifically, SOPA estimates the conditional probability $\theta_{[j]}^l$ of $\mathcal{O}_{[j]}^l$ given the already decoded octants $\mathcal{O}_{[:j-1]}^l$ and latent $\hat{\mathbf{f}}^l$:

$$\theta_{[j]}^l = \Psi_j^l\left(\hat{\mathbf{f}}_{[j]}^l, \hat{\mathbf{f}}^l; \psi_j^l\right), \quad \hat{\mathbf{f}}_{[j]}^l = \Phi_j^l\left(\mathcal{O}_{[:j-1]}^l, \hat{\mathbf{f}}^l; \phi_j^l\right), \tag{4}$$

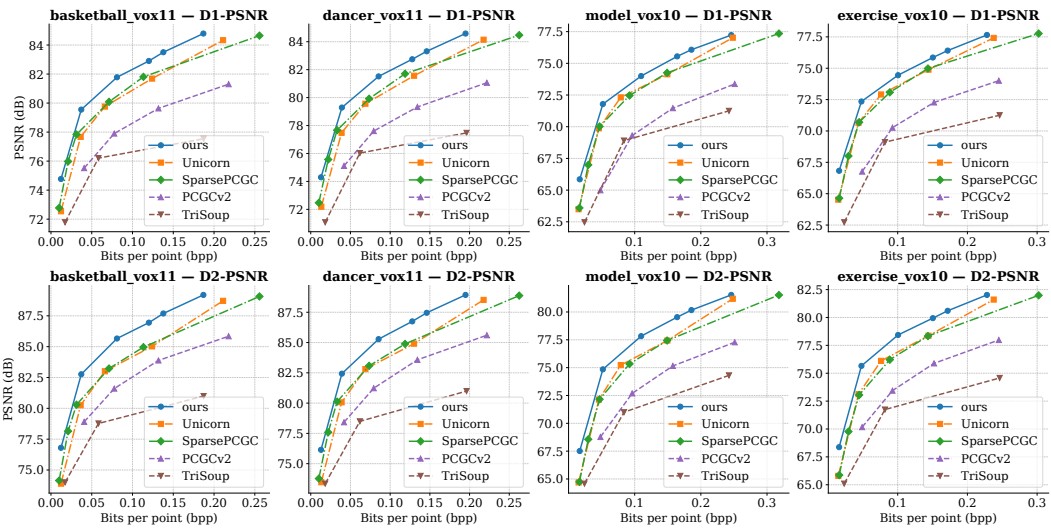

Figure 4: The comparison between SurfelSoup (ours) and baselines on Owlii.

where $\Psi_j^l/\Phi_j^l(\cdot)$ are sparse convolution networks for probability estimation/latent reconstruction, parameterized by $\psi_j^l/\phi_j^l$. $\hat{\mathbf{f}}_{[j]} = \left\{ \hat{\mathbf{f}}_{i,j}^l | \forall i \right\}$ is the union of group $j$'s reconstructed latents, which integrates the information of $\hat{\mathbf{f}}^l$ and already decoded context $\mathcal{O}_{[j]}^l$.

The number of bits for compressing the octant occupancy information of octree node $\mathbf{o}_i^l$ is calculated by the sum of octants' binary cross entropy:

$$\mathcal{B}(\mathbf{o}_i^l) = -\sum_{j=0}^{7} \left[ \mathcal{O}_{i,j}^l \log_2 \theta_{i,j}^l + (1 - \mathcal{O}_{i,j}^l) \log_2 \left( 1 - \theta_{i,j}^l \right) \right]. \tag{5}$$

As shown in Alg. 1, during training each node $\mathbf{u}_{i,j}^l$ is always divided rather than prematurely terminated as a surfel node. Consequently, convolutions on $\mathcal{O}_{[0:j-1]}^l$ may access neighbors that would have been pruned when evaluating, since their parents have already been classified as surfel nodes (See Appendix A.2 and Fig. 10). To overcome such information leakage, we propose P-SOPA, designed for inputs with probabilistic existence. As illustrated in Fig. 3, when coding Group $j$, we Bernoulli sample a binary mask $m_{[0:j-1]}^l$ from the existence probabilities of already decoded octant groups $\mathcal{O}_{[0:j-1]}^l$: $\mathcal{P}_{[0:j-1]}^l = \left\{ \mathcal{P}_{i,j'}^l = \tilde{p}_i^l | \forall i; j' \in [: j-1] \right\}$. In addition, to preserve gradient flow, we apply a straight-through estimator (STE) (Bengio et al., 2013):

$$\bar{m}_{[0:j-1]}^l = \text{sg}\big(m_{[0:j-1]}^l - \mathcal{P}_{[0:j-1]}^l\big) + \mathcal{P}_{[0:j-1]}^l, \tag{6}$$

where $\text{sg}(\cdot)$ denotes the stop-gradient operator. The STE mask $\bar{m}_{[0:j-1]}^l$ is then applied to SOPA's context extraction network on $\mathcal{O}_{[0:j-1]}^l$. More information of P-SOPA and its implementation is shown in Appendix A.2 and A.3. The whole algorithm of P-SOPA is shown in Alg. 2.

### 3.6 LOSS FUNCTION FORMULATED AS EXPECTATIONS

We adopt a rate–distortion loss to jointly optimize rate and reconstruction quality:

$$\mathcal{L} = \lambda \cdot \mathcal{D} + \mathcal{R}_f + \mathcal{R}_o, \tag{7}$$

where $\mathcal{R}_f$ denotes the bit rate of compressing latent $\mathbf{f}^L$ using the hyperprior entropy model (Ballé et al., 2018). Due to the probabilistic nature of the decision module (Sec. 3.4), both the distortion term $\mathcal{D}$ and the octree bit rate $\mathcal{R}_o$ are defined as expectations over the decision probabilities:

$$\mathcal{D} = \frac{1}{N} \sum_{l,i} \tilde{q}_i^l \mathcal{D}(\mathbf{s}_i^l), \quad \mathcal{R}_o = \frac{1}{N} \sum_{l,i} \tilde{p}_i^l \mathcal{B}(\mathbf{o}_i^l), \tag{8}$$

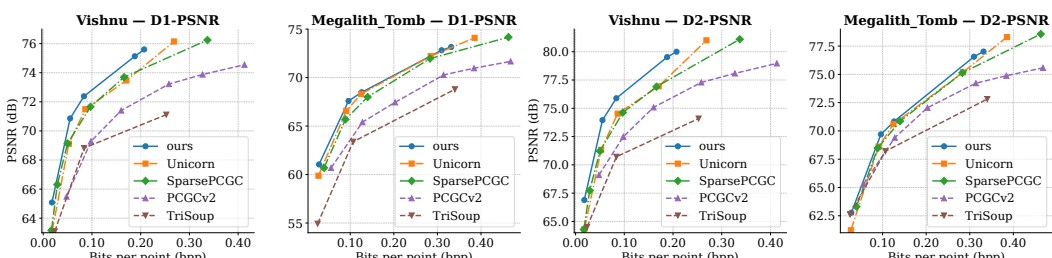

Figure 5: The comparison between SurfelSoup (ours) and baselines on RWTT dataset.

Table 1: BD-rate (%) comparison of our method against different baselines. Negative values indicate bitrate savings over the baseline.

| Baseline | Basketball | | Dancer | | Model | | Exercise | | Megalith Tomb | | Vishnu | |
|---|---|---|---|---|---|---|---|---|---|---|---|---|
| | D1 | D2 | D1 | D2 | D1 | D2 | D1 | D2 | D1 | D2 | D1 | D2 |
| Unicorn | -35.81 | -38.73 | -34.08 | -36.15 | -24.50 | -28.81 | -24.17 | -28.97 | -13.13 | -13.60 | -28.27 | -32.85 |
| SparsePCGC | -28.87 | -34.23 | -26.41 | -31.00 | -28.18 | -33.53 | -28.32 | -34.12 | -24.03 | -14.88 | -26.09 | -33.44 |
| PCGCv2 | -69.37 | -61.99 | -68.42 | -60.51 | -69.41 | -63.52 | -68.71 | -65.43 | -51.28 | -28.55 | -60.24 | -55.98 |
| TriSoup | -74.98 | -73.98 | -73.62 | -72.82 | -68.44 | -69.55 | -74.14 | -71.50 | -62.95 | -29.10 | -57.28 | -63.57 |

where $\mathcal{D}(\mathbf{s}_i^l)$ is the surfel distortion in Eq. 2, $\mathcal{B}(\mathbf{o}_i^l)$ is the octree node occupancy bits in Eq. 5, $\tilde{p}_i^l$ and $\tilde{q}_i^l$ denotes the probability of a node being an octree node or a surfel node, given in Eq. 3.

# 4 EXPERIMENTS

## 4.1 DATASETS

We train our model on the training dataset that the MPEG Common Test Condition (CTC) specifies: 8i Voxelized Full Bodies (8iVFB) (d'Eon et al., 2017). We use the five sequences in 8iVFB for training, including *longdress*, *redandblack*, *soldier*, *loot* and *queen*. We evaluate our model and baselines on the CTC-specified testing dataset: Owlii (Keming et al., 2018). We use the four sequences in Owlii for evaluation, including two 10-bit sequences *exercise* and *model*; and two 11-bit sequences *basketball_player* and *dancer*. We put the MPEG CTC's 12-bit sequence *ThaiDancer* in Appendix B.4. To evaluate the generalization ability of our model, we also evaluate on the object and scene dataset: Real World Texture Things (RWTT) (Maggiordomo et al., 2020).

## 4.2 EXPERIMENT SETUP

**Baselines.** We compare with: Unicorn (Wang et al., 2024), SparsePCGC (Wang et al., 2022) and PCGCv2 (Wang et al., 2021a) retrained on the same 8iVFB dataset as our method, which show better performance under MPEG CTC test condition compared with the original version trained on ShapeNet (Chang et al., 2015). We obtain the results from authors of (Xu et al., 2025). We also compare with MPEG G-PCC-TriSoup (Zhang et al., 2024) using the G-PCC-GesTM-TriSoup software, which outperforms G-PCC-Octree on dense point clouds. We did not compare with UniPCGC (Wang & Gao, 2025) because its Variable Rate and Complexity Module (VRCM) is designed for fine-grained rate control rather than for improving geometry modeling, which can be integrated to all the baselines and our method.

**Evaluation Metrics.** We evaluate the quality of the point cloud geometry reconstruction by D1-PSNR (point to point) and D2-PSNR (point to plane) calculated through the MPEG *pc_error*.

**Model Configuration.** The number of down-sampling stages $L$ is set to 3. The geometry coordinates up to layer $L = 3$ is losslessly coded with G-PCC-Octree. Since encoding the octants' occupancy information of nodes at $l = 3$ requires an extremely low bitrate ($< 0.015$ bpp), performing G-Surfel reconstruction at this level becomes suboptimal under the rate–distortion trade-off. Therefore, nodes at $l = 3$ are forced to be classified as octree nodes during training/evaluation. Furthermore, given the excessive complexity and bit rate needed to code the occupancy information at layer $l = 0$, nodes at $l = 1$ are forced to be classified as surfel nodes.

Figure 6: Visual comparison of decoded point clouds by SurfelSoup vs. baselines. The colors of decoded points are interpolated from colors of the original point cloud.

## 4.3 EXPERIMENT RESULTS

**Owlii.** We first evaluate our model on the Owlii dataset. The rate–distortion (RD) curves are shown in Fig.4, while the BD-rate results are summarized in Tab.1. Leveraging the proposed G-Surfel representation, which compactly models smooth point cloud surfaces, our method achieves an average BD-rate reduction of $-29.64\%$ (D1) and $-33.17\%$ (D2) over Unicorn, the current state-of-the-art method for dense point cloud geometry compression (Xu et al., 2025). SurfelSoup also achieves consistent gain compared with other baselines. Notably, the improvements are more significant on the two 11-bit (vox11) sequences, which exhibit more smooth surfaces that are easily fitted by G-Surfels.

**RWTT.** To further validate the generalization ability of our model trained on human data only, we evaluate **without finetuning** on the RWTT dataset, specifically the two sequences *Megalith Tomb* and *Vishnu*, which are also included in MPEG's test set. The RD-curves are shown in Fig. 5. Our method outperforms the other baselines. The gain on *Megalith Tomb* is less significant than on *Vishnu*, as *Megalith Tomb* is a large-scale scene point cloud with irregular geometry, whereas *Vishnu* is an object point cloud that contains more smooth surface structures.

**Visual Comparison.** We compare the rendered views of compressed point clouds using OpenGL (Zhou et al., 2018) in Fig. 6. Voxel-based baselines suffer from the discontinuity of voxel representations. This manifests especially under low bitrates due to **error propagation**: when coarse octree layers are lossy coded, dropping a single occupied octant removes all the corresponding fine-level voxels, leading to large gaps. This issue is especially pronounced on flat surfaces, where the resulting holes and cracks become highly visible. In contrast, SurfelSoup models the whole node as a continuous surface element, which exibits no error propagation and preserves smooth and coherent geometry even under low bit rate.

## 4.4 ABLATION STUDIES

**Decision Module.** We test SurfelSoup without the decision module. As shown in Fig. 7, forcing the G-Surfel Nodes to terminate at $l = 1$ only ($2 \times 2 \times 2$ cubes) or at $l = 2$ only ($4 \times 4 \times 4$ cubes) leads to inferior performance at low or high bitrates, respectively.

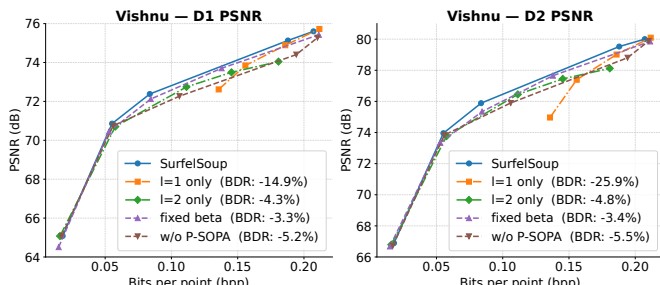

Figure 7: Ablation studies done on RWTT-Vishnu. "$l = 1$ only" and "$l = 2$ only" denotes removing the G-SurfelTree decision and constructs G-Surfels on only one layer. "fixed beta" denotes fixing the G-Surfel shape coefficient $\beta = 2$. "w/o P-SOPA" denotes replacing P-SOPA with SOPA.

**P-SOPA.** We test SurfelSoup trained without probabilistic masking in P-SOPA. As shown in Fig. 7, the model has a huge performance drop in middle rate points, where the surfel nodes spread in different octree layers. Under that case, conducting SOPA on a node during training will access neighbors that should have been pruned during evaluation (See example in Appendix A.2 and Fig. 10), because their parents are classified as surfel nodes. Therefore, the information leakage during training is not negligible, leading to a mismatch between training and evaluation.

**Shape coefficient $\beta$.** We retrain by fixing $\beta = 2$. As shown in Fig. 7, we can witness a consistent performance drop when replacing the generalized Gaussian function by the Gaussian function.

## 4.5 G-SURFEL PARAMETER DISTRIBUTION

We visualize the histogram of the variance $\{\sigma_i | i \in \{0, 1, 2\}\}$ and shape coefficient $\beta$ in Fig. 8.

**Variance:** At octree layer $l = 2$, there are typically two axes with large variances ($\sigma_0$ and $\sigma_1$) and one with significantly smaller variance ($\sigma_2$). For quantification, we define the planarity ratio $r = \min(\sigma_0, \sigma_1)/\sigma_2$, where $\sigma_2$ corresponds to the smallest variance (surfel thickness along normal). A high $r$ indicates that the G-Surfel is effectively constrained to a thin surface. At $l = 2$, the ratio $r$ is consistently large, confirming that $l = 2$ G-Surfels behave as planar structures bounded by the octree box. In contrast, at $l = 1$, the variances across the three dimensions are more balanced, leading to smaller values of $r$. This suggests that the G-Surfels at $l = 1$ tend to approximate Gaussian-like blobs rather than thin planes, enabling the capture of non-planar local structures.

**Shape coefficient $\beta$:** $\beta$ exhibits a wider distribution at $l = 1$. At this resolution, $\beta$ plays a stronger role in controlling the drop-off sharpness of occupancy distribution, leading to greater variability. In contrast, at $l = 2$, the G-Surfels are coarser and often extend beyond the bounding box, making the influence of $\beta$ within the bounded region less significant.

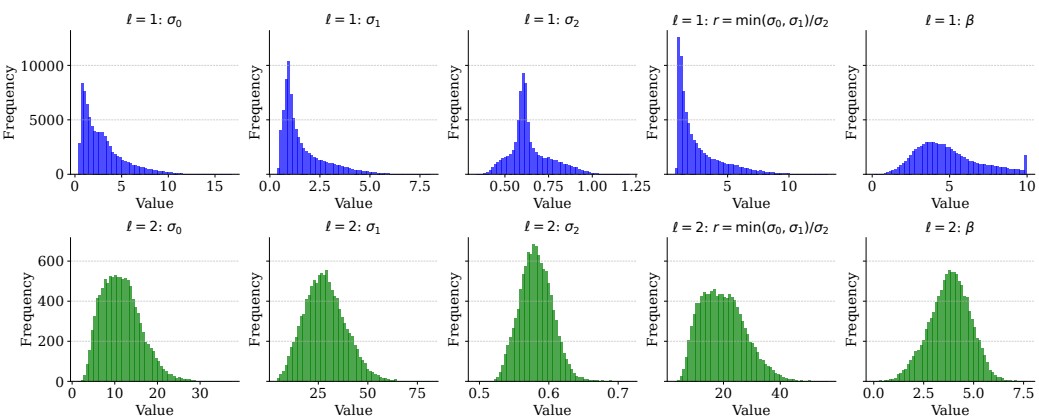

Figure 8: Distribution of G-Surfel Parameters.

Figure 9: Visualization of different layers of decoded point clouds. It is noted that coarse layers capture flat surfaces (base of sculpture, ground), while fine layers capture detailed information (sculpture, trees).

### 4.6 VISUALIZATION OF DIFFERENT LAYERS

Fig. 9 visualizes the point cloud reconstructed from G-Surfels at different layers of G-SurfelTree. We pick a middle rate point corresponding to $\lambda = 0.8$, where around half of the surfel nodes terminates at $l = 2$ and the remaining at $l = 1$. We can observe that G-Surfels at coarser layers capture the smooth surfaces (*e.g.* the ground). G-Surfels at finer layers usually represent high-frequency details with uneven surface normals (*e.g.* trees). This indicates that the decision module learns to distinguish point cloud slices with different geometry complexities and adaptively selects the surfel granularity to reach rate-distortion optimality.

### 4.7 COMPLEXITY

**Encoding/Decoding Time.**   Despite the introduction of G-Surfel, our encoding/decoding time is almost identical compared with Unicorn and SparsePCGC, because the bulk of time is on SOPA's auto-regressive coding of octree nodes.

**Model Size.**   Our model has larger size compared with Unicorn due to the introduction of G-Surfel Reconstruction and decision module. However, these models are all constructed by stride-one convolutions (See Appendix A.3) to avoid the information leakage mentioned in Sec. 4.4. Therefore, the overall model size is only moderately larger than that of Unicorn. In addition, the model size can be further reduced before real applications, with the pruning, distillation and quantization of model parameters (See Appendix B.6).

Table 2: Comparison of Method Complexities.

| Metric | Method | | | | |
|---|---|---|---|---|---|
| | SurfelSoup | Unicorn | SparsePCGC | PCGCv2 | TriSoup |
| Enc Time (s) | 1.1 | 1.1 | 1.2 | 0.4 | 2.8 |
| Dec Time (s) | 1.4 | 1.2 | 1.3 | 0.5 | 1.1 |
| Model Size (MB) | 33 | 22 | 12 | 4 | 0 |

## 5 CONCLUSION

This paper proposes the first end-to-end learned surface-based point cloud geometry compression framework. It proposes a novel probabilistic surface representation termed G-Surfel for efficient dense geometry compression and compact yet smooth rendering. We further introduce a G-SurfelTree structure assigning G-surfels to different tree levels to reach optimal trade-off between rate and distortion. Experiments show significant improvement over voxel-based baselines. Our future work will extend this work for color compression. In addition, we plan to explore SurfelSoup's extension to sparse point clouds without explicit surface structures.

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

## A  APPENDIX FOR METHODS

### A.1  NOTATIONS

We summarize the main notation used throughout the paper in Tab. 3. For better understanding, we further provide an illustration of each notation in Fig. 10.

Table 3: Summary of Notation.

| Symbol | Description |
|---|---|
| $L$ | Maximum number of downsampling in feature encoding (set as 3) |
| $l \in \{0, \cdots, L\}$ | Octree level index, 0 denotes the finest level, $L$ the coarsest level. |
| $\mathbf{u}_i^l$ | A node $i$ at level $l$ |
| $\mathbf{u}_{i,j}^l$ | The $j$-th octant of node $\mathbf{u}_i^l$ |
| $\hat{\mathbf{f}}_i^l$ | Latent feature of $\mathbf{u}_i^l$ |
| $\hat{\mathbf{f}}_{i,j}^l$ | Latent feature of octant $\mathbf{u}_{i,j}^l$ |
| $\mathbf{s}_i^l$ | $\mathbf{u}_i^l$ classfied as a surfel node |
| $\mathbf{o}_i^l$ | $\mathbf{u}_i^l$ classfied as an octree node |
| $\mathcal{V}_i^l$ | The set of voxels $\mathbf{v}_{i,k}^l$ in the finest level in the bounding box defined by $\mathbf{s}_i^l$ |
| $\mu, \sigma, q, \beta$ | Mean, variance, quaternion and shape of G-Surfels |
| $\mathcal{D}(\mathbf{s}_i^l)$ | Distortion incurred by representing $\mathbf{u}_i^l$ as a surfel |
| $\pi(i, l)$ | The index of the parent of $\mathbf{u}_i^l$ in layer $l+1$ |
| $\tilde{p}_i^l$ | Probability of $\mathbf{u}_i^l$ being an octree node estimated by the decision module |
| $\tilde{q}_i^l$ | Probability of $\mathbf{u}_i^l$ being a surfel node estimated by the decision module |
| $\mathcal{O}_{i,j}^l \in \{0, 1\}$ | Ground truth occupancy of $j$-th octant under node $i$, $\mathbf{u}_{i,j}^l$ |
| $\mathcal{O}_i^l \in \{0,1\}^8$ | Occupancy of 8 octants under node $\mathbf{u}_i^l$, $\mathcal{O}_i^l = \{\mathcal{O}_{i,j}^l | j \in [:8]\}$ |
| $\mathcal{O}_{[j]}^l$ | The set of octants in layer $l$ with group index $j$, $\mathcal{O}_{[j]}^l = \{\mathcal{O}_{i,j}^l | \forall i\}$ |
| $\mathcal{O}_{[:j-1]}^l$ | The set of octants in layer $l$ with group index $j' < j$, $\mathcal{O}_{[:j-1]}^l = \{\mathcal{O}_{i,j} | \forall i; j' \in [:j-1]\}$ |
| $\theta_{i,j}^l$ | P-SOPA-predicted probability that $\mathcal{O}_{i,j}^l = 1$ |
| $\theta_{[j]}^l$ | P-SOPA-predicted probabilities of group $j$ octants, $\theta_{[j]}^l = \{\theta_{i,j}^l | \forall i\}$ |
| $\mathcal{P}_{[:j-1]}^l$ | Probability of existence of $\mathcal{O}_{[:j-1]}^l$, $\mathcal{P}_{[:j-1]}^l = \{\mathcal{P}_{i,j'}^l = \tilde{p}_i^l | \forall i; j' \in [:j-1]\}$ |
| $m_{[:j-1]}^l$ | Bernoulli sampled mask based on $\mathcal{P}_{[:j-1]}^l$ |
| $\bar{m}_{[:j-1]}^l$ | STE version of $m_{[:j-1]}^l$ |
| $\mathcal{B}(\mathbf{o}_i^l)$ | Bit rate estimation of coding the occupancy of octants of $\mathbf{o}_i^l$, $\mathcal{O}_i^l$ |
| $\epsilon$ | Threshold for surfel termination during evaluation |
| Bernoulli$(\cdot)$ | Bernoulli distribution |

## A.2 More Information on P-SOPA

We give a toy example about why we care about the information leakage during training in the middle of Fig. 10, with explanations why information leakage occur in the caption. Therefore, to make training more close to real decoding, we apply a Bernoulli-STE mask $\bar{m}_{[:j-1]}^l$ on the already decoded octants $\mathcal{O}_{[:j-1]}^l$, to simulate the process during evaluation that some octants in $\mathcal{O}_{[:j-1]}^l$ do not exist, because their parents are classified as surfel nodes (which are no further divided). This mask, $\bar{m}_{[:j-1]}^l$, is generated by duplicating the parents' probability of being classified as octree nodes, $i.e.$ $\tilde{p}^l$. Because octants exist iff. their parents are octree nodes that need further division.

## A.3 Network Structures

This section introduces the network architecture of SurfelSoup, including the encoder module (Sec. 3.2), G-Surfel Reconstruction (Sec. 3.3), decision module (Sec. 3.4) and P-SOPA (Sec. 3.5).

Figure 10: (a) Illustrations of notations. (b) a toy example of information leakage during training. The red dashed box indicates convolution, only four octants out of the eight octants are drawn. Black/white/red indicate that an octant is occupied/unoccupied/unknown (not decoded yet). The decoding of groups is conducted from left to right. When decoding the last group of $\mathbf{u}_i^l$, we apply a convolution (red dashed line) on the nearby already decoded groups for context generation. However, during evaluation, some decoded octants may not exist (*e.g.* $\mathbf{u}_{i'}^l$'s first node) because its parent is classified as a surfel ($\mathbf{s}_{i'}^l$), while during training the network can still access it because no actual termination is performed.(c) a toy example with only two nodes of P-SOPA. The color of $\tilde{p}^l$ and $\mathcal{P}_{[:j-1]}^l$ indicates the value of probability, with darker color indicating higher probability of existence. We sample a Bernoulli-STE mask $\bar{m}_{[:j-1]}^l$ from $\mathcal{P}_{[:j-1]}^l$ to prune the already decoded octants, so that the training better simulates the actual decoding.

### A.3.1 ENCODER

The encoder architecture is shown in Fig. 11 (a). We follow the encoder design in SparsePCGC (Wang et al., 2022) and Unicorn (Wang et al., 2024), consisting of two ResNet-3 Modules (Wang et al., 2024) and one stride-two sparse convolution layer for downsampling.

### A.3.2 P-SOPA

The detailed architecture of P-SOPA is shown in Fig. 11 (c). P-SOPA follows the design of the SOPA module in SparsePCGC (Wang et al., 2022). However, the ResNet-3 is replaced by P-ResNet-3, where an STE mask in Sec. 4.4 is applied to the input of every convolution layer.

### A.3.3 DECISION/G-SURFEL RECONSTRUCTOR

The decision module/G-Surfel Reconstructor is shown in Fig. 11 (b), with one ResNet-1 Module and a stride-one sparse convolution module that changes the output to the desired dimension. For decision module, there is an additional Gumbel-Softmax. Note that we use stride-one instead of stride-three convolutions to avoid the information leakage mentioned in Sec. 3.5.

**G-Surfel parameters.** A G-Surfel is defined by its mean $\mu \in \mathbb{R}^3$, variance $\sigma \in \mathbb{R}^3$, quaternion $q \in \mathbb{R}^4$ and shape coefficient $\beta \in \mathbb{R}^3$. The rotation matrix is derived by $q = [w, x, y, z]$:

$$R(q) = \begin{bmatrix} w^2 + x^2 - y^2 - z^2 & 2(xy - wz) & 2(xz + wy) \\ 2(xy + wz) & w^2 - x^2 + y^2 - z^2 & 2(yz - wx) \\ 2(xz - wy) & 2(yz + wx) & w^2 - x^2 - y^2 + z^2 \end{bmatrix}. \tag{9}$$

The covariance matrix is derived by the rotation matrix $R(q)$ and variance $\sigma$:

$$\Sigma = R \operatorname{diag}(\sigma_0^2, \sigma_1^2, \sigma_2^2) R^\top, \qquad \varepsilon = \Sigma^{-1} = R \operatorname{diag}(\sigma_0^{-2}, \sigma_1^{-2}, \sigma_2^{-2}) R^\top. \tag{10}$$

### A.4 COMPARISON WITH OTHER 3D REPRESENTATION

It is worth clarifying how SurfelSoup differs from recent 3D representation techniques such as Gaussian splatting, neural implicit fields, signed distance functions, and surfels. These methods aim to

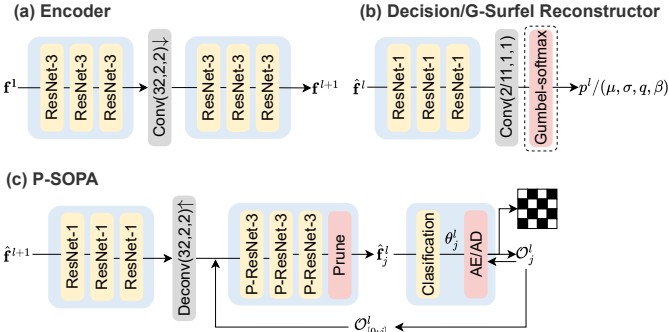

Figure 11: The architecture for (a) Encoder, (b) Decision Module/G-Surfel Reconstructor and (c) P-SOPA. ResNet-$x$ denotes the Sparse ResNet structure (Wang et al., 2024) constructed by stride-$x$ convolutions. Conv($C$,$K$,$S$) denotes convolution with channel size $C$, kernel size $K$ and stride $S$. The dashed-line box indicates that Gumbel-Softmax is only for decision module. *AE/AD* denotes the arithmetic encoder and arithmetic decoder.

build continuous or differentiable scene representations from multi-view supervision and per-scene optimization. These methods often targets at the quality of novel-view rendering, and often have huge amount of parameters (>20MB) because no rate constraint is applied. In contrast, our setting focuses on point cloud geometry compression, where the goal is to encode the input 3D point cloud into a compact bitstream (<0.05MB) under standardized rate–distortion constraints, while ensuring the quality of the 3D reconstructed point cloud. For concreteness, we provide a qualitative comparison with 3D Gaussian Splatting in Tab. 4, while the same mismatch in assumptions and objectives applies broadly to other 3D representation families.

Table 4: Comparison between SurfelSoup and 3D Gaussian Splatting (3DGS).

| **Property** | **SurfelSoup** | **3DGS** |
|---|---|---|
| Representation primitives | Probabilistic G-Surfels | 3D Gaussians |
| Structure | Hierarchical G-SurfelTree | unstructured set of Gaussians |
| Training paradigm | Dataset-level learning | Per-scene optimization |
| Objective | Rate–distortion optimization | Photometric rendering loss |
| Representation target | Geometry only | Geometry + color + opacity |
| Compressibility | Built-in entropy coding | External compression required |
| Granularity control | Learned adaptive decision | Hand-crafted prune/densify |
| Scene generalization | Yes | No |
| Primary use-case | Point cloud geometry compression | 3D reconstruction for rendering |
| Speed | Instant | Per-scene optimization |

## B APPENDIX FOR EXPERIMENTS

### B.1 SURFEL RENDERING.

In Fig. 12, we compare rendered images by three methods: OpenGL point cloud rendering for SurfelSoup reconstructed points, Our bounded 2D surfel based rendering for surfelSoup directly, and OpenGL point cloud rendering for Unicorn.

**OpenGL Rendering.** In OpenGL (Zhou et al., 2018) rendering, each point is splatted on to the screen using a fixed $n \times n$ pixel patch, where $n$ is manually set. To ensure fairness, we determine $n$ by selecting the minimum number that eliminates visible gaps in the rendered images from the

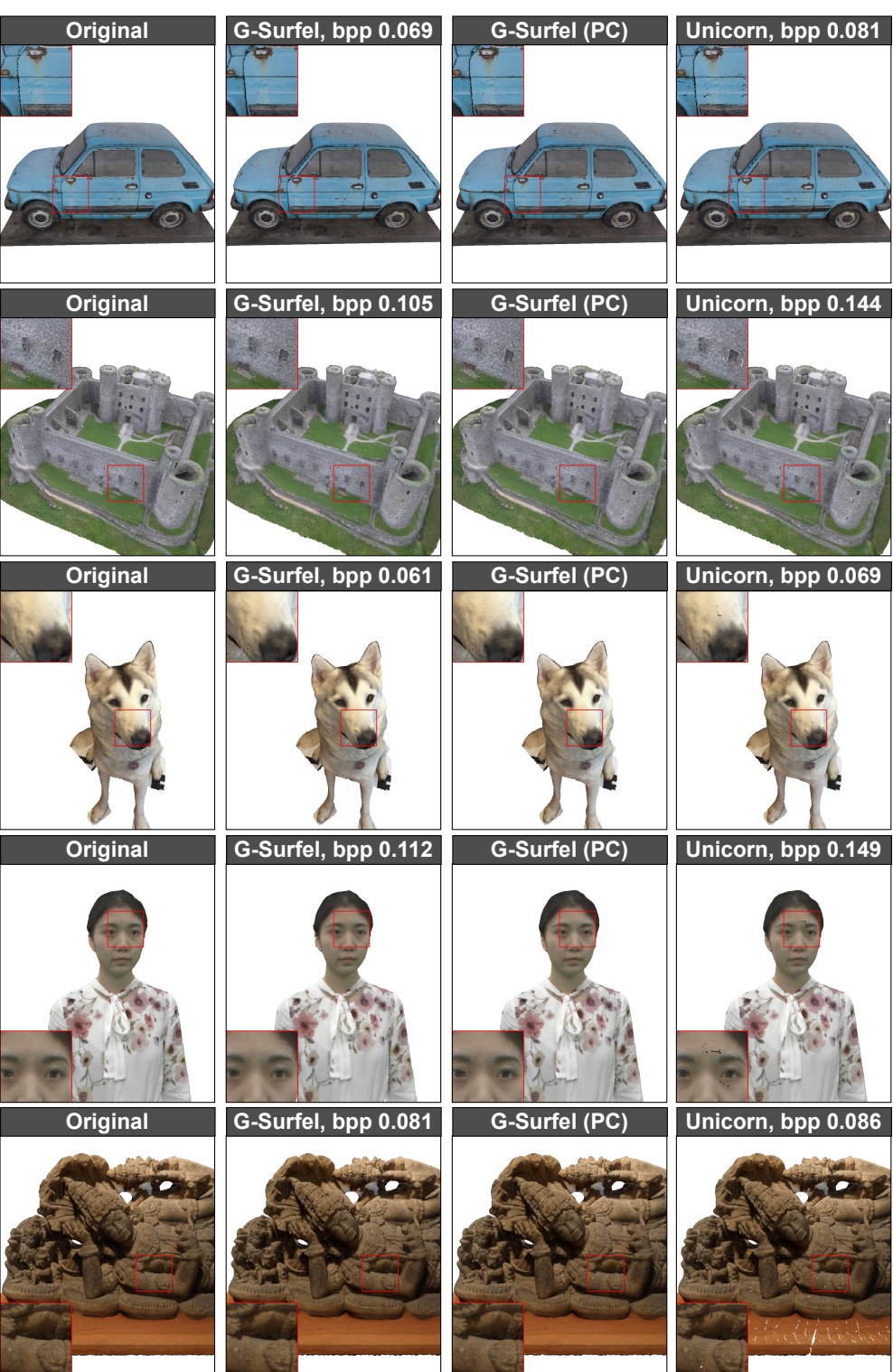

Figure 12: Comparison between rendered images from the decoded point clouds using different methods. Column G-Surfel denotes visualization of decoded surfels by SurfelSoup. Column G-Surfel (PC) denotes visualization of decoded points by SurfelSoup using OpenGL. Unicorn indicates the rendered images of decoded points by Unicorn using OpenGL. The color of a decoded point is obtained by interpolating on the colors of the original point cloud. Row 1, 2 and 3 are three samples from RWTT (Maggiordomo et al., 2020).

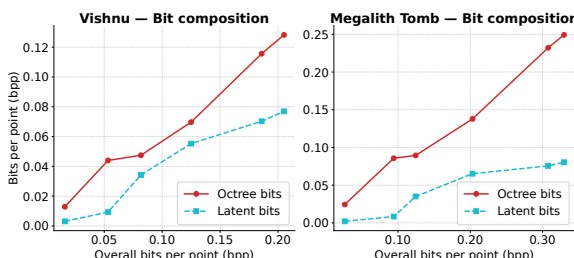

Figure 13: Bit rate composition (octree bits and latent bits).

ground-truth uncompressed point cloud. This chosen point size is then applied to both G-Surfel (PC) and Unicorn.

**Surfel Rendering**   In addition to rendering the decoded point clouds by SurfelSoup using OpenGL, we can also render the decoded surfels directly. For each cube decoded to a G-Surfel, we identify the direction with the minimal variance and use it as the normal direction of the surfel plane. We define the two axes of the surfel by a rotation aligning the global $z$-axis with the normal. We construct an $n \times n$ texture map on this tangent plane, where $n$ varies with the octree level, so that it covers the largest possible surfel bounded by the cube. All decoded points within the cube and its neighbors are projected to this plane, which ensures consistency of texture samples across neighboring cubes. For each pixel in the texture map, we query the k-nearest projected points and assign the pixel a color equal to the inverse-distance-weighted average of these k neighbors. We use $k = 3$, $n = 7$ for $l = 1$, $n = 15$ for $l = 2$.

For rendering, we perform rasterization from the given camera view. We first sort all potential surfels for a pixel by their distances to the camera center. Then, along each camera ray, we compute the intersection point between the ray and the surfel surface. If this point lies inside the surfel's bounding box, we convert this point into the tangent plane coordinate and query the corresponding texture map using bilinear interpolation to shade the pixel from the texture patch. Otherwise, we proceed to the next surfel until a valid hit is found. This design enables efficient CUDA-based rasterization, follows the logic of textured 2D Gaussians(Chao et al., 2025), but leverages a bounded surfel representations(Hu et al., 2025).

**Rendering Result**   We can observe that at similar geometry rates, both G-Surfel and G-Surfel (PC) significantly outperform Unicorn, which suffer from visible gaps in some areas. Compared with the G-Surfel renders using OpenGL, surfel rendering provides slightly more smooth color and geometry distribution (the readers may need to further zoom in).

Due to the different convergence speeds of different octree layer's G-Surfel Reconstruction (finer layer converges faster because it has less points inside each surfel node), the network cannot be trained from scratch. Otherwise, the decision module converges to a local minimum, which classifies all the nodes in coarser layers as octree nodes. Instead, we first pre-train a model by setting $\tilde{q}_i^l = 0.5$ and $\lambda = 2.0$ to better support the convergence of different layers' G-Surfel Reconstruction. We then fine-tune this pre-trained model with different $\lambda$.

For Gumbel Softmax, the temperature coefficient is set as $\tau = 1.5 \times \exp(-0.0002 \times \text{step})$.

### B.2   BITS COMPOSITION

We visualize the bit rate composition (bpp for octree compression and bpp for latent compression) in Fig. 13. Compared with voxel baselines (Wang et al., 2021a; 2024; 2022) which codes the entire octree to some level, our method supports terminating a portion of the octree as surfels through the decision module. This gives the network great flexibility in adaptively selecting data structures (voxels or surfels) for geometric representation.

We can further observe that: 1) the octree takes more bits as total rate goes higher, indicating that more nodes in coarser layers are represented as octree nodes for better quality surfels in higher layers; and 2) compare with Megalith Tomb, the percentage of octree bits of Vishnu is lower, indicating

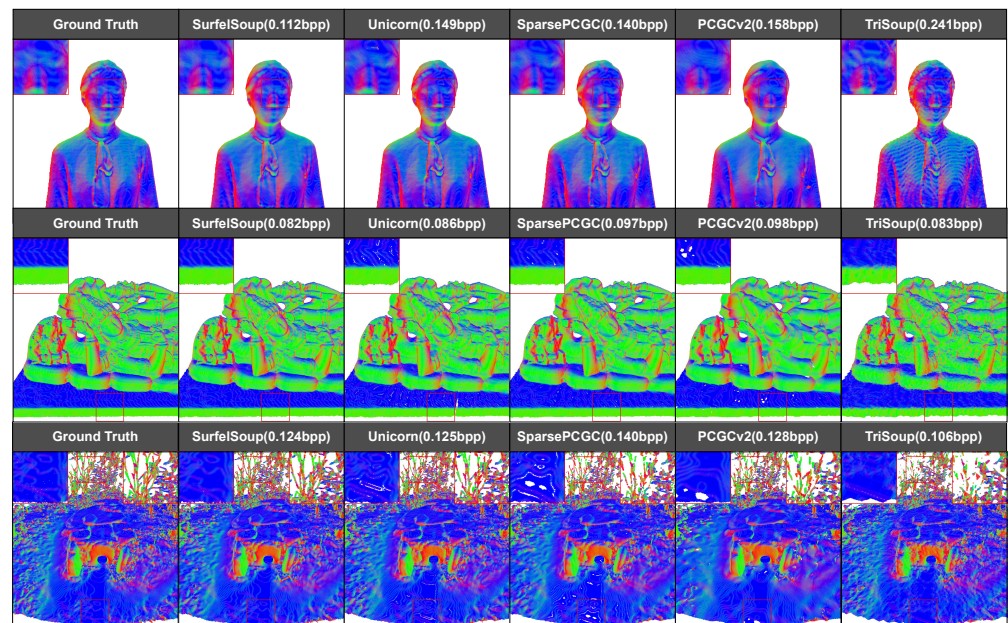

Figure 14: Visual comparison between SurfelSoup and baselines. The point color is visualized by the normal vector.

that Vishnu has more surface structures. This aligns with the fact that Megalith Tomb is an irregular scene point cloud, while Vishnu is a dense object point cloud.

## B.3 VISUAL COMPARISON

We compare the visualization between SurfelSoup and baselines in Fig. 14. Different from Fig. 6, the visualization in this section renders no color information. Because human eyes are more sensitive to distortion when color is rendered, we put the pure geometry visualization in the Appendix.

## B.4 EXTRA RESULTS

**ThaiDancer:** To further demonstrate the generalizability of our models trained on the 10-bit 8iVFB dataset, Fig. 15 shows the testing result on the 12-bit MPEG CTC test sequence *ThaiDancer_vox12*.

**8iVFB:** To provide test result on 8iVFB, we retrain our model on another MPEG CTC-specified training set: RWTT (Maggiordomo et al., 2020). The result is provided in Fig. 16. We compare with SparsePCGC by retraining SparsePCGC also on RWTT. We do not compare with Unicorn because Unicorn is not open-source yet, thus we cannot retrain their model. We do not compare with PCGCv2 because its performance is shown to be worse than Unicorn and SparsePCGC in previous studies. In addition, we compare with MPEG standard G-PCC-TriSoup.

## B.5 TRAINING DETAILS

We train five models with $\lambda = 0.1, 0.3, 0.8, 1.0, 1.5$. The initial learning rate is set as $0.0001$, which is multiplied by $0.85$ after every 15 epochs. The model for each rate point is trained for one day. The lowest $\lambda$ ($\lambda = 0.1$) corresponds to bpp $\approx 0.05$. To obtain even lower rate points, we adopt the super-resolution approach in SparsePCGC (Wang et al., 2022) and Unicorn (Wang et al., 2024). Specifically we downsample the original point cloud by a stride-2 pooling layer, and compress this down-sampled point cloud by our trained model with $\lambda = 0.1$. We then leverage a super-resolution network Wang et al. (2024) to upsample the decoded point cloud to the original scale.

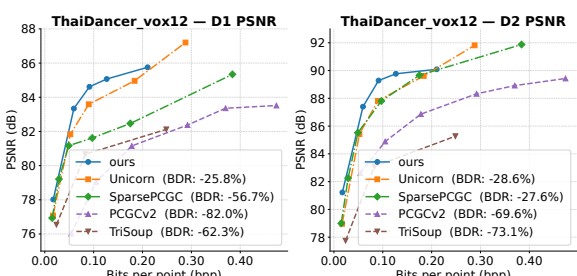

Figure 15: Comparison of SurfelSoup and baselines on *ThaiDancer_vox12*.

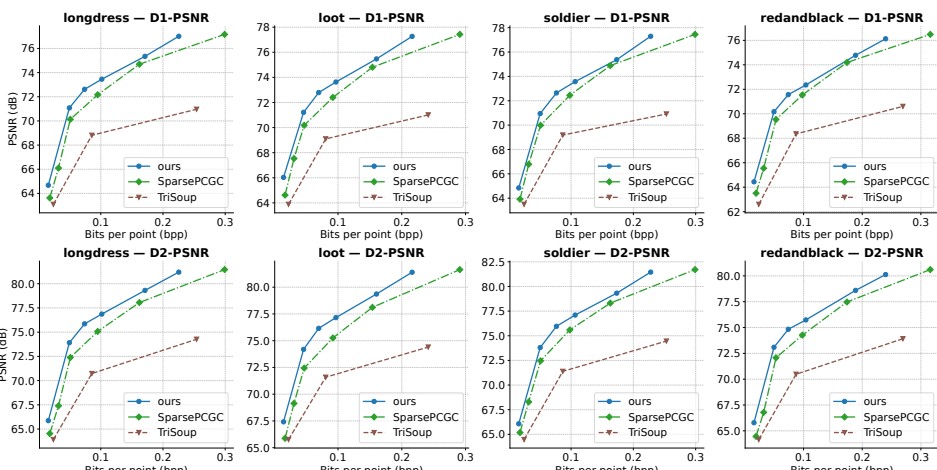

Figure 16: Comparison of SurfelSoup and baselines on 8iVFB. SurfelSoup and SparsePCGC are retrained on RWTT.

## B.6 LIMITATIONS AND FUTURE WORK

**Scene Point Clouds:** SurfelSoup shows significant gain against voxel-based baselines on dense point clouds with smooth surface structures. However, the gain on structurally complex scenes is limited because most surface areas need to be divided to the finest layer $l = 1$. The potential improvements include:

- **Coordinate system:** SCP (Luo et al., 2024) provides one way of preprocessing sparse LiDAR point clouds by transfering the xyz coordinate system into spherical coordinate system for more regularized point clouds. Experimental result shows around 30% gain by simply changing the coordinate system. SurfelSoup can follow the same pipeline to make the scene point cloud denser, and more surface-like.

- **Dataset** SurfelSoup actually has the potential of dealing with sparse point clouds due to the flexibility of G-Surfels. We have mentioned in Sec. 3.3 that when the variance $\sigma \in \mathbb{R}^3$ is large in two axis and small in the other one, the produced G-Surfel approximates a surface structure. On the other hand, when $\sigma$ is small in all the three axis, the produced G-Surfel can represent tiny and irregular geometry structures, which is widely observed in sparse point clouds. Under extreme case when $\sigma \to 0$, G-Surfel represents a single point defined by the center $\mu$, which can perfectly depict the isolated points in sparse point clouds. Therefore, SurfelSoup may be greatly improved on sparse point clouds, if it can be trained on sparse datasets like SemanticKITTI.

**Color:** SurfelSoup currently does not support attribute (color) compression. However, SurfelSoup can be extended to attribute compression by 1) adding a texture map representation for each surfel as shown in Appendix B.1; or 2) adding a separate color attribute $\mathbf{c} \in \mathbb{R}^3$ to G-Surfel parameters. Note that a flat surface may have a complex texture pattern. Therefore, we may need multiple single-colored surfels to represent a flat surface area.

**Model Size:**  SurfelSoup's model size is 33MB. Although this is substantially smaller than widely-used vision backbones such as VGG-16 ($\sim$528 MB) or ResNet-50 ($\sim$98 MB), the model size can be further reduced by model pruning, model distillation and model quantization. Classic "Deep Compression" (Han et al., 2016) pipelines have shown up to 49× reduction without sacrificing accuracy on vision backbones. Recent learned image compression models demonstrate that distillation (Chen et al., 2025) can reduce parameters by about 40%.

## C    TECHNICAL CLARIFICATION NOTE

**Intuition behind G-Surfel as Primitive**  : A G-Surfel models a local point set as a small surface patch parameterized by a generalized Gaussian (GG) function. The key intuitions are as follows:

- Surface (surfel) is a more compact representation than voxels/points, which can represent a local point set lying on the same underlying surface. This leads to both structural compactness and reconstruction smoothness.
- The measuring of point-to-surfel distance is ambiguous. Naively using point-to-plane distance (D2) restrcits nothing about surfel thickness and size. Therefore, we turn to the probabilistic representation, with a 3D GG function differentiably depicting the occupancy probability of each voxel.

## D    REPRODUCIBILITY STATEMENT

**Code Release:**  We plan to publicly release the full implementation, including training and evaluation scripts and model weights upon acceptance of this paper.

**Datasets:**  Our experiments use publicly available datasets under MPEG Common Test Condition (CTC). Note that RWTT (Maggiordomo et al., 2020) is a mesh dataset, where we use the script from MPEG for point cloud sampling and quantization.

**Compute Resource:**  The training is done on single NVIDIA A100 GPU for one day, identical to the training time of baselines. The peak memory consumption during training is 23 GB.

