# OpenReview forum: "SurfelSoup: Probabilistic G-SurfelTree for Learned Point Cloud Geometry Compression"
_ICLR.cc/2026/Conference — Submitted to ICLR 2026_

### Official Review · Reviewer_yRia · 2025-10-26

**Soundness:** 3
**Presentation:** 2
**Contribution:** 2
**Rating:** 4
**Confidence:** 3

**Summary:**

The paper presents a contribution by introducing SurfelSoup, an end-to-end surface-based framework for point cloud compression, which breaks away from the traditional voxel-based paradigm.  It utilizes a probabilistic surface representation called G-Surfel, modeling voxel occupancies within an octree node using a bounded 3D generalized Gaussian function, thereby facilitating the use of a differentiable distortion term in the loss function during training.  Another significant aspect is the proposed G-SurfelTree hierarchy, which enables adaptive G-Surfel granularity assignment across octree levels via a decision module and a corresponding formulation of the total expected rate and distortion, rather than relying on a predefined tree structure.

**Strengths:**

* The paper shows high technical rationality by introducing an end-to-end surface-based point cloud compression framework.
* The paper provides a clear and detailed explanation of the proposed method.
* The paper conducts a thorough experimental comparison with several representative methods.

**Weaknesses:**

* Certain statements in the paper are ambiguous. For instance, the first surface-based framework lacks well-defined boundaries. The definition of a surface-based approach is unclear, especially since previous methods using surface triangle information and occupancy (a surface-oriented representation) cast doubt on their surface-based classification. Also, although the proposed model shares similarities with 3DGS, the paper doesn't provide an in-depth and positive discussion on their differences.
* Limited Visual and Quantitative Analysis. In the visual comparison, only the comparison between the proposed method and Unicorn is shown. The authors should present the visual effects of all methods involved in the quantitative analysis and render color-free geometries to show geometric structure compression precision.
* The quantitative comparison reveals that the performance improvement of the proposed method is relatively limited.

**Questions:**

* The first surface-based framework raises some ambiguous boundaries. What exactly constitutes a surface-based approach? Previous methods have utilized surface triangle information. Do they fall under the category of surface-based? Occupancy, in essence, is also a surface-oriented representation method. Does it qualify as surface-based?
* Section 2.3 only discusses one single paper. In fact, Surfel is a fundamental geometric concept, and many research studies on 3D representation have proposed various modeling methods for surfels. The introduction in the paper is rather one-sided and incomplete.
* In the visual comparison, the paper only shows the comparison between the proposed method and Unicorn. The authors should present the visual effects of all methods involved in the quantitative analysis. * Moreover, they should render the color-free geometries to demonstrate the compression precision of geometric structures.
* There are discrete artifacts and splitting phenomena in Unicorn. The authors should conduct a more in-depth discussion on why these situations occur.

---

> ### Author Response · Authors · 2025-11-18
> **Response to reviewer yRia**
>
> Thank you for your recognition in our design rationale, writing and experiments. Our response to each of the weakness are show below:
>
> **Weakness 1 \& Question 1**:
>   1) We would like to clarify our claim regarding being the first **end-to-end** surface-based point cloud compression framework. Indeed, some prior methods do leverage surface information—for example, TriSoup uses triangular surface patches and TeSO employs cubic bounded planar  surfaces. However, these approaches are not end-to-end learned codecs. They rely on hand-crafted surface extraction algorithms followed by separate entropy coding, without differentiable rate–distortion optimization or learned surface primitives. This is fundamentally different from our formulation.
>
>   2) We add a qualitative comparison with 3D Gaussian Spatting methods in Appendix A.4. It shows SurfelSoup's difference with 3D GS in the aspects of objective, structure, scene generalization, use case, etc.
>
>   3) We also acknowledge the reviewer’s concern about the ambiguity between “surface-based” and “surface-oriented”.
> To remove this ambiguity, we have clarified in the paper that SurfelSoup is surface-based in the sense that explicit surfaces are treated as primitives of geometry representation. This differentiates our approach from voxel-occupancy based methods which only implicitly optimize towards surface structures.
>
> **Weakness 2, Question 3**:
> In Fig. 6, we add the visual comparison with all baselines mentioned in quantitative comparison. In Appendix B.3 and Fig. 14, we add the pure geometry visualization (color represent the  normal vectors).
>
> **Weakness 3**:
> We would like to clarify that SurfelSoup achieves approximately a -30\% BD-Rate improvement over the previous voxel-based baselines Unicorn that is already highly optimized (Tab. 1).
> This indicates that, for the same reconstruction quality, SurfelSoup requires about 30% fewer bits. In recent MPEG standardization documents and related literature, a -5–10% BD-Rate improvement is typically regarded as a  significant advancement.
>
> **Question 2**:
> We substantially revised Section 2.3 to provide a more comprehensive overview of 3D surface representation related works. We now include both implicit representations [1] [2] [3] [4] and explicit representations [5] [6] [7] [8].
>
> Notably, surfel [5] proposes to represent a surface with isolated surface elements without storing the connectivity, hence achieving efficient rendering and processing flexibility. Later works introduce the surface splatting [9] [10] techniques and non-square shapes [11] to address the visibility issues in rendering. By designing the primitives with differentiability, surfels and more generally 3D Gaussians have been shown to facilitate 3D reconstruction from multi-view images and novel view [12] [6]. If there are still important papers about 3D surface representation that we missed, we would appreciate your letting us know.
>
> **Question 4**:
> We add a detailed explanation of why there are discrete artifacts and splitting phenomena in voxel baselines in Section 4.3 (visual comparison). Voxel-based baselines suffer from the discontinuity of voxel representations. This manifests especially under low bitrates due to **error propagation**: when coarse octree layers are lossy coded, dropping a single occupied octant removes all the corresponding fine-level voxels, leading to large gaps. This issue is especially pronounced on flat surfaces, where the resulting holes and cracks are highly visible.
>
> [1] Learning continuous signed distance functions for shape representation. CVPR 2019.
>
> [2] Plenoxels: Radiance fields without neural networks. CVPR 2022.
>
> [3] Nerf: Representing scenes as neural radiance fields for view synthesis, Communications of the ACM 2021.
>
> [4] Neural unsigned distance fields for implicit function learning. NeurIPS, 2020
>
> [5] Surfels: Surface elements as rendering primitives. In Proceedings of the 27th annual conference on Computer graphics and interactive techniques
>
> [6] High-quality surface reconstruction using gaussian surfels. ACM SIGGRAPH 2024
>
> [7] Surfelnerf: Neural surfel radiance fields for online photorealistic reconstruction of indoor scenes. CVPR 2023
>
> [8] Neural point-based graphics. ECCV 2020
>
> [9] Surface splatting. Proceedings of the 28th annual conference on Computer graphics and interactive techniques.
>
> [10] A survey of point-based techniques in computer graphics. Computers & Graphics
>
> [11] High-quality surface splatting on today's GPUs. Proceedings Eurographics/IEEE VGTC Symposium Point-Based Graphics, 2005.
>
> [12] 3D Gaussian splatting for real-time radiance field rendering. ACM Trans. Graph.

---

### Official Review · Reviewer_jDDz · 2025-10-29

**Soundness:** 3
**Presentation:** 3
**Contribution:** 2
**Rating:** 4
**Confidence:** 4

**Summary:**

This paper presents  a novel end-to-end learned framework for dense point cloud geometry compression. Unlike previous voxel-based approaches such as PCGCv2, SparsePCGC, and Unicorn, SurfelSoup models 3D geometry as a composition of probabilistic surfaces (G-Surfels) rather than discrete voxels. Each G-Surfel is formulated as a bounded generalized Gaussian distribution, parameterized by a mean, covariance, quaternion rotation, and shape coefficient, which can flexibly represent local planar or curved surface patches. The method further organizes these G-Surfels hierarchically in a G-SurfelTree, analogous to an adaptive octree, where a decision module learns whether to subdivide or terminate a node based on rate-distortion optimization. During inference, the model reconstructs point clouds by binarizing occupancy probabilities, selecting voxels with the highest likelihood. Experiments on several benchmarks demonstrate that SurfelSoup achieves significantly higher compression efficiency than Unicorn and other state-of-the-art baselines while producing smoother and gap-free reconstructed surfaces.

**Strengths:**

The introduction of the probabilistic G-Surfel is conceptually elegant and bridges voxel-based compression with continuous surface modeling, similar in spirit to 3D Gaussian Splatting. The probabilistic surface modeling avoids voxelization artifacts, leading to visually continuous and realistic point clouds.

The entire framework is end-to-end.

**Weaknesses:**

This method reconstructs the point cloud based on resampling, so the reconstructed point cloud is different from the original one. Moreover, this method can only compress the geometry and cannot be used to compress the RGB colors, which reduces its value and makes the motivation less clear.

The paper still lacks a comparison with 3DGS compression methods. Essentially, both approaches aim to compress point clouds, but 3DGS compression methods also compress the color information of the scene.

**Questions:**

The proposed method can only compress geometry, so how were the colors in the images shown in the experiments generated?

---

> ### Author Response · Authors · 2025-11-18
> **Response to reviewer jDDz**
>
> Thank you for your recognition in our conceptual design and visualization results. Our detailed responses to each weakness are provided below:
>
> **Weakness 1**: Point cloud compression can be divided into lossy and lossless compression, where lossy compression is our focus. Quantitative comparison on MPEG-specified D1- and D2-PSNR, as well as the visualization have proven that SurfelSoup's lossy compression saves a lot of bits without hurting the quality. Besides, in the extreme case, lossless coding can be achieved by arithmetically coding the occupancy using the predicted probability.
>
> In addition, MPEG (V-PCC and G-PCC) suggests dividing point cloud compression into geometry and attribute compression, respectively, which is also followed by research on AI-based point cloud compression. Attribute compression is usually decoupled and separate from geometry compression. It first recolors the decompressed geometry points based on the ground truth, then compresses the recolored colors. As we stated in our title, our paper and baselines SparsePCGC and PCGCv2 are for pure geometry compression. While Unicorn uses completely different networks for geometry [1] and attribute [2] compression. **We add discussion of  SurfelSoup's potential in color compression in Appendix B.6**. However, this paper focuses on geometry compression, and the joint compression of geometry and attribute is considered as future work.
>
> **Weakness 2**: We add a qualitative comparison between our SurfelSoup and 3D GS in Appendix A.4. To the best of our knowledge, there is no existing work using 3D GS for point cloud compression.
>
> A direct quantitative comparison with 3D GS compression is unfortunately not meaningful, for two fundamental reasons:
>   1) **Input**: 3D GS compression methods compress the 3D Gaussian primitives from multi-views instead of a point cloud.
>   2) **Evaluation Metric and Reconstruction**: SurfelSoup reconstructs point clouds with accuracy measured by 3D metrics (D1- and D2-PSNR); while 3D GS compression reconstructs rendered views measured by 2D metrics (MSE, SSIM and LPIPS).
>
> **Question 1**: The color is generated by recoloring based on the ground truth point cloud: For every point $\hat{p}$ inside the reconstructed point cloud $\hat{\bf p}$, we find its 3-nearest neighbors {$q_k|k=1,2,3$} in the ground truth point cloud. And the reconstructed color is generated by inverse-weighted interpolation of these 3 points. Because it is difficult for human eyes to perceive distortion when only geometry is shown, we instead compare illustrate recolored point clouds. We also provide pure-geometry comparison in Appendix B.3 by showing the normal maps.
>
> [1] A versatile point cloud compressor using universal multiscale conditional coding--part I: Geometry. TPAMI
>
> [2] A versatile point cloud compressor using universal multiscale conditional coding--part II: Attribute. TPAMI

---

### Official Review · Reviewer_7KPz · 2025-11-01

**Soundness:** 3
**Presentation:** 2
**Contribution:** 3
**Rating:** 4
**Confidence:** 3

**Summary:**

This paper proposes a novel surface-based point cloud geometry compression method called SurfelSoup, which for the first time combines a probabilistic surface representation (G-Surfel) with a learnable tree structure (G-SurfelTree), achieving end-to-end training and optimization. The method demonstrates superior performance on multiple standard datasets, particularly excelling in smooth surface reconstruction. The paper has a clear motivation, innovative methodology, and comprehensive experimental design, contributing significantly both theoretically and practically.

**Strengths:**

1. It introduces the first end-to-end learned surface-based point cloud compression framework, incorporating a probabilistic G-Surfel representation based on a generalized Gaussian distribution, making surface modeling more expressive and differentiable, demonstrating strong innovation.

2. The method is comprehensively designed, covering modules such as encoding, surface reconstruction, decision-making, and entropy coding, while addressing information leakage issues during both training and inference.

3. Extensive comparisons with various existing methods on standard test sets are provided, along with detailed ablation studies that validate the effectiveness of the proposed approach.

**Weaknesses:**

1. Some formulas and process descriptions in Section 3 (Methodology), particularly the probabilistic modeling of P-SOPA and the decision module, are somewhat obscure and could benefit from more intuitive diagrams or pseudocode for clarification.

2. The method primarily targets dense point clouds and shows limited performance on structurally complex scenes, indicating relatively high application constraints. It is recommended to analyze the model's adaptability to irregular geometric structures and propose potential improvements.

3. Although the experimental comparisons are extensive, they could be further strengthened by including comparisons with recent implicit or explicit surface reconstruction methods.

4. The increased model size compared to voxel-based baselines somewhat hinders practical deployment potential, warranting discussion on potential model compression or pruning strategies.

**Questions:**

1. Some formulas and process descriptions in Section 3 (Methodology), particularly the probabilistic modeling of P-SOPA and the decision module, are somewhat obscure and could benefit from more intuitive diagrams or pseudocode for clarification.

2. The method primarily targets dense point clouds and shows limited performance on structurally complex scenes, indicating relatively high application constraints. It is recommended to analyze the model's adaptability to irregular geometric structures and propose potential improvements.

3. Although the experimental comparisons are extensive, they could be further strengthened by including comparisons with recent implicit or explicit surface reconstruction methods.

4. The increased model size compared to voxel-based baselines somewhat hinders practical deployment potential, warranting discussion on potential model compression or pruning strategies.

---

> ### Author Response · Authors · 2025-11-18
> **Response to reviewer 7KPz**
>
> Thank you for your recognition in our novelty, design and experimental results. Our detailed response to each weakness is provided below:
>
> **Weakness 1 & Question 1**:
> To make our paper more readable,
>   1) we provide a notation table in Appendix A.1, Table 3.
>   2) We add one diagram illustrating all the notations, and introducing information leakage and how P-SOPA works in Fig. 10.
>   3) We add pseudocodes for both the decision module (Algorithm 1) and P-SOPA module (Algorithm 2), and we have substantially revised Sec. 3.4 (decision module) and 3.5 (P-SOPA) to aid the understanding of the pseudocodes.
>
> **Weakness 2 & Question 2**:
> We add Appendix B.6 to describe SurfelSoup's adaptability to sparse point clouds with irregular geometric structures.
>   1) **Coordinate system**: For sparse point clouds, previous work [1] has shown they can be transferred from xyz to spherical coordinate system. The transformed point cloud will have denser and more surface-like geometry structures.
>   2) **Retraining on sparse datasets**: SurfelSoup has the potential of dealing with sparse point clouds and may benefit from retraining on sparse datasets, like SementicKITTI. We have mentioned in Sec. 3.3 that when the variance $\sigma\in R^3$ is large in two axes and small in the other one, the produced G-Surfel approximates a surface structure. On the other hand, when $\sigma$ is small in all three axes, the produced G-Surfel can represent tiny and irregular geometry structures, which is widely observed in sparse point clouds. Under extreme case when $\sigma\rightarrow0$, G-Surfel represents a single point defined by the center $\mu$, which can perfectly depict the isolated points in sparse point clouds.
>
> **Weakness 3 & Question 3**:
>  We add the introduction of implicit [2] [3] [4] [5] and explicit surface representations [6] [7] [8] [9]) in Related work 2.3. If there are still important papers about 3D surface representation, we would appreciate your letting us know. While our method is inspired by these surface-based representations, a direct quantitative comparison is unfortunately not meaningful, for two fundamental reasons (**which is also introduced in Appendix A.4**):
>
>  1) **Uncompressed model**: Because the bit rate for describing surface parameters is not taken into account, these methods usually generates a large number of parameters and requires very large storage  (generally $>$10MB). On the other hand, by using a rate-distortion loss function together with a built-in entropy model for parameter compression, SurfelSoup  generally needs $<$0.1MB.
>
>  2) **Objectives**: Except for the early surfel works, these methods are designed for reconstructing 3D structure from multi-views and for novel view rendering. Such methods  cannot be compared directly to our method, which assume the point cloud as a 3D structure is given and focuses on the compression of the point cloud.
>
> The only  method that considers point cloud compression while leveraging a surface representation  is G-PCC-Trisoup, which uses a set of triangles to represent geometry. We have included a comparison with TriSoup in the experiment section.
>
> **Weakness 4 & Question 4**:
> We add a complexity discussion in Appendix B.6. SurfelSoup's model size is 33MB. This is 11MB larger than Unicorn, but it is already substantially smaller than widely-used vision backbones such as VGG-16 (528 MB) or ResNet-50 (98 MB). In addition, although model pruning, quantization, or distillation is not within this paper's scope, similar to most learning-based codecs, SurfelSoup can be further compressed for practical deployment, if desired. e.g., recent learned image compression models [10] demonstrate that distillation can reduce parameters by about 40%.
>
> [1] Scp: Spherical-coordinate-based learned point cloud compression. AAAI 2024.
>
> [2] Learning continuous signed distance functions for shape representation. CVPR 2019.
>
> [3] Plenoxels: Radiance fields without neural networks. CVPR 2022.
>
> [4] Nerf: Representing scenes as neural radiance fields for view synthesis, Communications of the ACM 2021.
>
> [5] Neural unsigned distance fields for implicit function learning. NeurIPS, 2020
>
> [6] Surfels: Surface elements as rendering primitives. In Proceedings of the 27th annual conference on Computer graphics and interactive techniques
>
> [7] High-quality surface reconstruction using gaussian surfels. ACM SIGGRAPH 2024
>
> [8] Surfelnerf: Neural surfel radiance fields for online photorealistic reconstruction of indoor scenes. CVPR 2023
>
> [9] Neural point-based graphics. ECCV 2020
>
> [10] Knowledge distillation for learned image compression. ICCV 2025

---

### Official Review · Reviewer_x6xe · 2025-11-09

**Soundness:** 3
**Presentation:** 3
**Contribution:** 2
**Rating:** 4
**Confidence:** 5

**Summary:**

This paper proposes SurfelSoup, the first end-to-end learned surface-based framework for point cloud geometry compression. It introduces a probabilistic and differentiable surface representation, termed G-Surfel, which models local point occupancies using a bounded generalized Gaussian distribution. A G-SurfelTree is then constructed as an adaptive octree-like hierarchy, where a decision module determines subdivision depth to achieve rate–distortion–optimal granularity.

**Strengths:**

The idea of this paper is motivated by the limitations of voxel-based learned point cloud compression methods. Then, a proposed G-surfel is used to reconstruct the point cloud from a coarse octree with the associated features.  The performance indicates the superiority of the proposed technique to some prior works.

**Weaknesses:**

1. This paper introduces a decision module to adaptively determine whether to terminate a node as a G-Surfel or further subdivide it within the G-SurfelTree. This component is claimed to be critical for achieving a rate–distortion–optimal balance and for adaptively allocating surfel granularity. But the notation in this section is not clear and needs improvement for better comprehension.
2. The bit rate of the octree compression should be analyzed.
3. The experimental results show that the proposed method outperforms Unicorn. However, it is unclear which components contribute most to this improvement.
4. High complexity in terms of model size, encoding, and decoding time.

**Questions:**

1. The model configuration in section 4.2 is difficult to catch, particularly with the notation l and L.
2. The training dataset is different from most of the prior works. Besides, 8i VFB is usually used as a test dataset for the static point cloud.

---

> ### Author Response · Authors · 2025-11-18
> **Response to Reviewer x6xe**
>
> Thank you for your recognition of our motivation and performance. Our detailed responses to each weakness are provided below.
>
> **Weakness 1**:
> To make our paper more readable,
>   1) we provide a notation table in Appendix A.1, Table 3.
>   2) We add one figure illustrating all the notations in Figure 10.
>   3) We add pseudocodes for both the decision module (Algorithm 1) and P-SOPA module (Algorithm 2), and we have substantially revised  Sections 3.4 and 3.5  to better explain the algorithms along with the pseudocodes.
>
> **Weakness 2**:
> The bit rate of octree occupancy compression and latent compression is shown in Figure 13. The corresponding analysis is in Appendix B.2. We also provide the table version of result here:
> |              Vishnu     | RP1        | RP2        | RP3        | RP4        | RP5        | RP6        |
> |-----------------|------------|------------|------------|------------|------------|------------|
> | **Total bpp**   | 0.01595    | 0.05323    | 0.08167    | 0.12495    | 0.18590    | 0.20519    |
> | **Octree bpp**  | 0.01286    | 0.04397    | 0.04744    | 0.06968    | 0.11565    | 0.12826    |
> | **Latent bpp**  | 0.00309    | 0.00927    | 0.03423    | 0.05527    | 0.07025    | 0.07693    |
>
> | Megalith Tomb       | RP1        | RP2        | RP3        | RP4        | RP5        | RP6        |
> |-----------------|------------|------------|------------|------------|------------|------------|
> | **Total bpp**   | 0.02643    | 0.09409    | 0.12443    | 0.20307    | 0.30769    | 0.32959    |
> | **Octree bpp**  | 0.02437    | 0.08571    | 0.08940    | 0.13787    | 0.23214    | 0.24923    |
> | **Latent bpp**  | 0.00206    | 0.00839    | 0.03502    | 0.06520    | 0.07555    | 0.08036    |
>
> We can observe that:
>   1) the octree takes more bits as total rate goes higher, indicating more surfels are coded at fine layers for better quality.
>   2) the percentage of octree bits of object point clouds (Vishnu) is lower, indicating more surface in object point clouds.
>
> **Weakness 3**:
> Compared with Unicorn, the key idea in the paper is to represent a large area of smooth surface using a group of surface elements, i.e. G-Surfel, which uses only 11 surface parameters to define a local smooth surface that needs otherwise coding many individual points to characterize. In addition, we have provided ablation studies in Figure 7 to show the effect of each component.
>
> **Weakness 4**:
> We add complexity discussion in Appendix B.6.
>
> We would like to kindly point out that our encoding time is the same as the other two top-performing learning methods and our decoding time is only slightly more  (0.2s$>$Unicorn and 0.1s$>$SparsePCGC).
>
> SurfelSoup's model size is 33MB, which is already substantially smaller than widely-used vision backbones such as VGG-16 (528 MB) or ResNet-50 (98 MB). In addition, although model pruning, quantization, or distillation is not within this paper's scope, similar to most learning-based codecs, SurfelSoup can be further compressed for practical deployment, if desired. e.g., recent learned image compression models [1] demonstrate that distillation can reduce parameters by about 40%.
>
> **Question 1**:
> We have provided the notation table and rewrote the model configuration section. $L$ denotes the maximum number of downsampling stage as well the layer where the latent features are entropy coded. $L$ is  set to 3 in our experiments. $l \in \{0,1,2,3\}$ refers to the octree level. For example, for a 10-bit point cloud, $l=0$ corresponds to 10 bits, $l=1$ to 9 bits, and so on. We will open source the implementation upon the acceptance of our paper, as we stated in Appendix C.
>
> **Question 2**:
> We follow the MPEG Common Test Condition [2] which specifies 8iVFB as training set and Owlii as test set.
>
> To see how SurfelSoup performs on 8iVFB, we retrain our model with another MPEG-CTC-specified dataset: RWTT. **We provide the result in Appendix B.4 (8iVFB) and Figure 16**. We compare with SparsePCGC by retraining SparsePCGC also on RWTT. Here we do not provide comparison to Unicorn because it is not open-source hence we cannot use the same training setting for a fair comparison. In addition, we compare with MPEG standard G-PCC-TriSoup. **SurfelSoup consistently outperformed SparsePCGC and G-PCC-TriSoup, as on the standard test condition**.
>
> | Methods (D1)       | Longdress        | Soldier        | Loot        | Redandblack        |
> |-----------------|------------|------------|------------|------------|
> | **SparsePCGC**   | -26.71    | -21.23    | -25.73   | -24.05    |
> | **TriSoup**  | -61.53    | -55.95    | -65.03    | -61.78    |
>
> [1] Yunuo Chen, Zezheng Lyu, Bing He, Ning Cao, Gang Chen, Guo Lu, and Wenjun Zhang. Knowledge distillation for learned image compression. In Proceedings of the IEEE/CVF International Conference on Computer Vision, pp. 4996–5006, 2025.
>
> [2] ISO/IEC JTC 1/SC 29/WG 7 MPEG 3D Graphics and Haptics Coding, CTC on AI-based point cloud coding

---

> > ### Author Response · Authors · 2025-11-29
> > **Response to Reviewer x6xe (Part II)**
> >
> > **Additional Response to Question 2**: D2 BD-rate on 8iVFB compared with SparsePCGC and TriSoup:
> >
> > | Methods (D2)       | Longdress        | Soldier        | Loot        | Redandblack        |
> > |-----------------|------------|------------|------------|------------|
> > | **SparsePCGC**   | -30.03    | -23.93    | -30.70   | -29.16   |
> > | **TriSoup**  | -65.27    | -59.51    | -64.08    | -64.38    |

---

### Author Response · Authors · 2025-11-29
**Summary of Our Response**

We appreciate the reviewers’ efforts and their constructive feedback. We carefully addressed all questions and substantially revised the manuscript to improve readability and technical soundness. Below is a concise summary of the major changes and clarifications made in response to the reviewers.

**Readability and Methodological Clarity**: To address concerns regarding notation, methodology, and model configuration clarity, we

  1) add a notation table (Appendix A.1).

  2) add pseudocode for the decision module and P-SOPA (Algorithms 1–2).

  3) add clearer diagrams illustrating the P-SOPA mechanism (Fig. 10, Appendix A.3).

  4) rewrite the model configuration section (Sec. 4.2 Model Configuration).

**Experiments**:
  1) add bit-rate composition analyses for latent and octree components (Fig. 13).

  2) add additional experiments by retraining on RWTT and testing on 8iVFB, including retraining SparsePCGC for fair comparison (Appendix B.4).

  3) update Fig. 6 to include visual comparisons with all baselines, and added pure-geometry visualizations (Appendix B.3).

**Complexity**: To address concerns that our model size (33 MB) is larger than the previous state-of-the-art Unicorn (22 MB), we

  1) add Appendix B.6 analyzing model size and complexity, which compares the model size with standard vision backbones, and discusses potential compression techniques such as pruning and distillation.

**Related Work and Comparison**: To address concerns regarding a comprehensive introduction of 3D explicit and implicit surface representations (especially 3D Gaussian Splatting and surfels) and a comparison with them, we

  1) rewrite related work in Section 2.3 to include explicit and implicit 3D surface representations, including Gaussian splatting, surfels, neural radiance fields, and signed distance functions.

  2) add a qualitative comparison with 3D Gaussian Splatting (Appendix A.4).

  3) explain in Appendix A.4 why we cannot do a quantitative comparison with 3D surface representation methods: because the proposed work is  targeted at 3D point cloud compression instead of 3D novel view rendering from given multiviews.

**Clarification of Performance Gains**: To clarify why SurfelSoup outperforms voxel-based baselines, and why voxel baselines show flawed 3D reconstructions, we

  1) explain in Sec. 4.3 that the gain comes from using surfels as primitives for compression, thereby avoiding the compression of individual voxels (points) by compressing the surface structure behind them. This is not only compression-efficient but also visual-friendly, because it retains the underlying surface structure and the reconstructed surfaces have less cracks and holes;

  2) add a paragraph in Sec. 4.3 (visual comparison) introducing the **error propagation** of voxel baselines, which causes the discrete artifacts of voxel baselines.

**Applicability**:  To address concerns regarding applicability to scene point clouds and attribute compression, we

  1) add Appendix B.6 to introduce how our method can be extended to scene point clouds and attribute compression.

  2) clarify that geometry-only compression aligns with the MPEG CTC setting.

**Terminology Clarification**: To address concerns about the definition of "surface-based", we

  1) slightly refine the phrasing around “surface-based.” Our original manuscript already stated that the contribution is the first **end-to-end** learned surface-based point cloud compression framework. The revised version further clarifies this meaning to avoid misinterpretation.

---

### Meta-Review · Area_Chair_oc3B · 2026-01-03

**Summary:**

This paper presents surface-based framework for point cloud compression. Unlike voxel-based approaches, this approach represents local point occupancies using the proposed G-Surfel, which is organized as a G-SurfelTree. The tree balances subdivision granularity to balance rate and distortion. The paper highlights that the method avoids redundant point-wise compression in smooth regions, helping to keep the compact representations. Experimental results consistently outperform voxel-based approaches (SparsePCGC, Unicorn) and the MPEG standard G-PCC-GesTM-TriSoup, yielding fewer cracks and holes.

**Reviewer Concerns:**

The reviewers noted the potential of the non-voxel-based approach but consistently raised the following concerns.
- Clarity (x6xe, 7KPz): In the method section, reviewers mentioned that the P-SOPA module is hard to follow and struggled with the notation.
- Computational complexity (x6xe, 7KPz): the encoding and decoding time is higher, and the model size is larger than that of the baseline Unicorn
- Insufficient comparison (jDDz, yRia, 7KPz): the reviewer requested additional comparison with 3DGS compression methods or requested comparison with explicit surface reconstruction methods.
- Limited applicability (7KPz, JDDz, yRia): the reviewers are concerned that the proposed method mainly targets dense point clouds, only handling geometry compression (while other approaches, such as Unicorn, can handle colors), and limited visual analysis.

**Reviewer Scores:**

This paper received the following review:
- x6xe: marginally below acceptance
- 7KPz: marginally below acceptance
- jDDz: marginally below acceptance
- yRia: marginally below acceptance

The reviewers reached a consensus with scores of 4.

The reviewer provided extensive revisions. They added a notation table, pseudocode, diagrams, new experiments on the RWTT dataset, visual comparisons, and a qualitative comparison with 3D Gaussian splatting.

However, AC confirms that the primary concerns: (1) higher model complexity, (2) the complexity of the formulation presents a barrier to reproducibility, (3) heavily optimized for dense point clouds, (4) comparison with relevant approaches (although the revision has a comparison with 3DGS, the advantage of G-Surfel over the emerging standards is not clear) are not fully addressed. Overall, AC confirms that the paper is not yet ready for publication at ICLR. However, authors are encouraged to improve the method by resolving (1) and (3) for a future submission.

---

### Decision · Program_Chairs · 2026-01-26

Reject